# Fewer Children in Families Associated with Lower Odds of Early Childhood Caries: A Sample from Three Countries

**DOI:** 10.3390/ijerph20032195

**Published:** 2023-01-26

**Authors:** Amy H. Park, Rachel J. Kulchar, Sita Manasa Susarla, Bathsheba Turton, Karen Sokal-Gutierrez

**Affiliations:** 1Rausser College of Natural Resources, University of California, Berkeley, CA 94704, USA; 2Department of Chemistry, Princeton University, Princeton, NJ 08544, USA; 3School of Public Health, University of California, Berkeley, CA 94704, USA; 4Henry M. Goldman School of Dental Medicine, Boston University, Boston, MA 02118, USA

**Keywords:** social determinants of health, child, oral health, dental caries, siblings, family characteristics

## Abstract

Childhood caries experience is influenced by family characteristics and oral health practices in the context of many social-commercial determinants. The aim of this study was to explore the relationship between families’ number of children, oral health practices and child caries experience in a convenience sample of 1374 children aged 6 months through 6 years and their families from Ecuador, Nepal, and Vietnam. Data were collected by mother interviews and child dental exams. Multivariate logistic and Zero-Inflated-Poisson regression analyses assessed associations between number of children, oral health practices and decayed, missing or filled teeth (dmft). Families had a mean of 2.2 children (range 1–12); 72% of children had tooth decay, with mean dmft of 5.4. Adjusting for child age, sex, and urban/rural location, a greater number of children in the family was associated with significantly less likelihood of unhealthy bottle feeding practices, having a toothbrush/toothpaste and parent helping child brush, and being cavity-free; higher number of dmft, and greater likelihood of having a dental visit. Early childhood oral health promotion should include focus on oral hygiene and healthy feeding—particularly breastfeeding and healthy bottle feeding practices—as well as access to family planning services and support for childcare.

## 1. Introduction

Market globalization and urbanization have contributed to a global nutrition transition from breastfeeding to bottle-feeding, and from traditional whole-food diets to ultra-processed foods and sugary beverages. This dietary shift has led to increasing rates of nutrition-related diseases in children and adults, including obesity, type 2 diabetes, cardiovascular disease, nonalcoholic fatty liver disease, cancers, and dental caries [1,2,3,4,5,6,7,8,9,10]. Among children, dental caries is the most prevalent chronic condition, affecting 60–90% of children globally [11]. For children in low-and middle-income countries (LMIC), and low-income populations in high-income countries, the high prevalence of dental caries and limited access to dental care has left much tooth decay untreated [12,13,14]. Untreated caries can contribute to a poorer quality of life, malnutrition, daily performance, and difficulty sleeping and eating due to mouth pain [4,15,16,17,18].

The development of early childhood caries (ECC)—tooth decay in children under age 6—is caused by a complex interaction among factors at the child, family, and societal levels [16,19,20,21,22]. At the child level, dietary sugars are metabolized by oral cariogenic bacteria to create acid which demineralizes the teeth, particularly with frequent and prolonged sugar exposure and inadequate toothbrushing habits [23,24,25]. Early childhood diet and oral health are shaped by many family factors as well as social and commercial determinants of health, including education and socioeconomic conditions, and marketing and access to healthy versus caries-causing foods and beverages, oral hygiene products, and dental care [26,27,28,29].

Family structure, including the number of children in the family, can influence many aspects of child health, including health practices and health outcomes [30,31]. Family structure is associated with the frequency of dental and medical treatment received [30], child obesity [31,32], malnutrition [33], and childhood mortality [34]. Systematic reviews have shown that larger family size, greater numbers of children, and being of higher birth order (i.e., having older siblings) are associated with greater risk of ECC [35,36]. Experts have hypothesized that an increased number of children in the family lessens the quality of home care and supervision that each child receives due to the “resource dilution model;”, i.e., distributing the limited parent/family finances, caregiving time and emotional resources among more children leaves less individual care available for each child [37,38]. 

To date, most studies have examined family size as a predictor of ECC without examining intermediary oral health practices that may be associated with family size and may contribute to ECC outcomes. The aim of this study was to utilize a dataset from a convenience sample of children and families from LMICs to explore the association between the number of children in a family, oral health practices, and children’s experience of ECC by their number of decayed, missing, or filled teeth (dmft).

## 2. Materials and Methods

### 2.1. Study Design and Population

This is a cross-sectional secondary analysis study of a convenience sample of 1374 low-income mothers and children aged 6 months through 6 years participating in a family of community-based programs promoting child nutrition and oral health in 3 LMICs: Ecuador, Nepal, and Vietnam. The communities were rural (Ecuador, Nepal), urban (Nepal, Vietnam), and peri-urban (Vietnam). This paper presents data collected from 2011 to 2013 in all three countries. Previous publications have detailed descriptions of the three sample sites and populations [4,39,40].

This family of studies was developed and conducted as a collaboration among University of California Berkeley (UCB) with local country Ministries of Health, academic institutions, and non-governmental non-profit organizations. All study sites and protocols were approved by the Committee for the Protection of Human Subjects, the Institutional Review Board (IRB) at UCB (#2010-06-1655, #2011-04-3176, #2011-04-3178). Additional approval was obtained, when required, by IRB reliance from the University of California, San Francisco, local ministries of health (Ecuador), health research council (Nepal), university (Vietnam), and non-profit organization directors (Ecuador, Vietnam, and Nepal).

### 2.2. Data Collection

In each site, trained local health workers explained the study to mothers in their native language, and obtained written and/or verbal informed consent for the mother’s and child’s participation. Consent forms and surveys were translated from English to the local languages and back translated to ensure accuracy. The survey was modified from the World Health Organization (WHO) oral health survey [41] for a low-literacy population, and consisted of 49 questions regarding maternal and family demographic characteristics, maternal-child nutrition and oral health knowledge and practices, complaint of oral symptoms, and maternal perception of her child’s oral health status. In Ecuador and Nepal trained local health workers and volunteers interviewed the mothers to complete the survey; and in Vietnam, the partner organization requested that the survey be completed directly by mothers since their education and literacy levels were high. Child dental exams were conducted by licensed local-country and United States (U.S.) dentists by visual inspection with a light and mirror, recording decayed, missing and filled teeth (dmft), and estimation of the depth of cavitation into the enamel, dentin or pulp. Exam results were calibrated through independent and joint examination of 5 children to establish agreement on findings. Previous publications have detailed descriptions of the three sites’ procedures for training and calibrating examiners [4,39,40].

After completing data collection, children received fluoride varnish application, and children and family members were provided oral health education and counseling individual toothbrushes and fluoride toothpaste, and referrals for dental treatment as needed.

### 2.3. Data Analysis

Data were recorded on paper forms and subsequently entered into Microsoft Excel Version 16.63.1 (Albuquerque, NM, USA). The data entered into Excel was systematically cleaned prior to performing data analysis. For this analysis, we included only the first visit of each mother with her oldest child under age 7 present, at the first visit. Missing data that were assumed to be random, could not be imputed and were ignored in the final analysis. 

We developed a Directed Acyclic Graph (DAG) to describe the hypothesized pathway that the number of children in the family is associated with maternal-child oral health practices and child oral health outcomes, and to identify potential confounders and covariates based on the literature (Figure 1). The questionnaire and exam data included continuous and categorical variables; some categorical variables were binary while others consisted of multiple categories that were converted to binary categories. Data were categorized into unhealthy baby bottle feeding practice if the child’s mean duration of bottle feeding was greater than 24 months, the child was bottle fed with sugary liquids, and/or the child was bottle fed while sleeping. Junk food consumption combined the consumption of sweets, chips/biscuits, and/or soda. 

Descriptive analysis, logistic regression, and Zero-Inflated Poisson (ZIP) regression models were performed through RStudio version 4.0.3 (PBC, Boston, MA, USA) [42]. G*power version 3.1.9.6 (Heinrich Heine University Düsseldorf, Düsseldorf, Germany) was used to calculate implied power [43]. To compare the families with one child and multiple children, the number of children in the family was made bivariate in the descriptive analysis. Data analysis with SPSS version 27 (IBM Corp, Armonk, NY, USA) yielded inferential analyses for independent samples to test for statistical significance between one child and multiple children. After adjusting for confounders and covariates (Figure 1), logistic regression and ZIP regression were performed to assess the association between number of children in the family and oral health practices and child caries outcomes (by dmft). Statistical significance was considered for *p*-values less than 0.05. The questionnaire did not inquire directly about family income to avoid shame and stigma, and based on literature showing that income is poorly reported in LMICs. Therefore, we used other proxies for socioeconomic status (SES) that have been validated to differentiate resource levels within low-income populations—including maternal education; household access to electricity, potable water, cooking fuel other than wood; urban vs. rural residence; and distance to a store. Urban vs. rural residence was selected as the single covariate to represent overall family SES to avoid collinearity among the aforementioned proxy measures in the regression model; and to account for maternal education and distance to a junk food store for which studies have demonstrated association with child tooth decay [44,45]. Sensitivity analyses were conducted to obtain a crude estimate of the relationship between number of children in the family and child’s caries outcome as well as identify the impact of covariates (Appendix A).

## 3. Results

### 3.1. Family Demographic Characteristics

In this sample, the children’s mean age was 3.7 years, 52.7% were male and 47.3% female. The mothers had a mean age of 30.9 years, and 8.7 years of education. Families had a mean of 2.2 children (range 1–12), with higher numbers of children in rural sites. Overall, 84.7% of the participants could walk from home to a store that sold junk food within 5 min (Table 1).

### 3.2. Oral Health Knowledge and Practices

There were discrepancies in knowledge and practice. While most mothers knew that eating sweets causes caries (73.6%), 29.9% of their children consumed sweets daily, 34.6% consumed any junk food or sugary drink daily, and only 6.9% of mothers restricted candy consumption. Less than 1% of mothers knew that using the baby bottle could cause ECC. Across all three countries, over 95% of children were breastfed, but over half of the children (52.8%) were also bottle fed, for a mean duration of 27.5 months; and 44.6% of mothers practiced unhealthy baby bottle feeding practices—29.4% of children were bottle fed with sugary liquids (including juice, soda, and sugar water), 14.6% of children were bottle fed to sleep, and 34.3% of children were bottle fed past 24 months of age. Bottle feeding practices ranged from 19% in Nepal to 40.1% in Ecuador to 86.1% in Vietnam where nearly two-thirds of children were bottle fed sugary liquids (62.3%) and longer than 24 months (62.8%). Only-children were significantly more likely to be bottle fed compared to children with siblings (64.2% vs. 46%). Nine out of ten children were reported to have their own toothbrush and access to toothpaste, but only half of children (54.6%) had parental help brushing their teeth, ranging from 24.5% in Nepal to 54% in Ecuador to 80% in Vietnam; and only-children were more likely than children with siblings to receive help brushing (64.2% vs. 48.9%). While many mothers knew complications of childhood caries—including pain (56.0%) and difficulty eating (40.9%)—only 38.1% of children had visited the dentist, ranging from 9.7% in Nepal to 44.5% in Vietnam to 74% in Ecuador; and only-children were less likely than children with siblings to have a dental visit (30.9% vs. 42%) (Table 2).

### 3.3. Child Oral Health Status

Overall, the child dental exams indicated that three out of four children (72.3%) had decayed, missing or filled teeth, with a mean dmft of 5.4; and one out of four children (28.5%) had deep decay into or near the pulp, by visual inspection. Only-children had less severe caries experience than children with multiple siblings (Mean dmft 4.4 vs. 6.0; Prevalence of deep decay 16.9% vs. 28%). Half the children (51.2%) experienced mouth pain, and a quarter of the children (22.9%) frequently or always experienced mouth pain. Mothers were over five-times as likely to assess their children’s oral health as “bad” (17.4%) compared to their general health (2.9%) (Table 3).

### 3.4. Association between Number of Children and Early Childhood Caries

After adjusting for confounders and covariates in the logistic regression analysis, a one-unit increase in the number of children was significantly associated with a lesser likelihood of unhealthy bottle feeding practices (aOR = 0.85), a lesser likelihood of a child having a toothbrush (aOR = 0.89) and toothpaste (aOR = 0.83), a lesser likelihood of the mother helping brush their child’s teeth (aOR = 0.86), a greater likelihood of a child visiting a dentist (aOR = 1.19), and a greater likelihood of a child having one or more carious lesions (aOR = 1.18). A one-unit increase in the number of children was not significantly associated with practices around consuming junk food daily or limiting sweets. 

After adjusting for covariates and confounders in the ZIP analysis, among children who have non-zero dmft (i.e., any decayed teeth), the baseline number of dmft is 4.57; and a one-unit increase in the number of children significantly increased the expected dmft count by 1.04 times. Among all children, a one-unit increase in the number of children significantly decreased the odds of being in the cavity-free group by a factor of 0.85. The implied power analysis suggested a 95% certainty that the observed findings were due to a real difference in the subgroups: has caries and caries-free (Table 4).

## 4. Discussion

This cross-sectional study of a convenience sample of children aged 6 months through 6 years across 3 LMICs demonstrated a high prevalence and severity of childhood caries, and widespread risk factors for caries, including high rates of unhealthy bottle feeding practices, proximity to stores that sell low-cost junk food, frequent consumption of junk food, limited help for children with toothbrushing, and limited dental care. There was a significant association between family size and oral health practices whereby a one-unit increase in the number of children in the family was associated with some healthier practices (less likelihood of bottle feeding with sugary liquids, bottle feeding in bed, and bottle feeding past 24 months of age), as well as some less-healthy practices (the child not having a toothbrush and toothpaste and not having parental help with brushing). Overall, a one-unit increase in the number of children in the family was significantly associated with a greater likelihood of a child having any decayed, missing or filled teeth (dmft), a greater number of decayed teeth among children with decay, and less likelihood of a child being cavity-free.

This study supports the findings of other studies showing high rates of caries and widespread dietary and oral hygiene risk factors, limited access to dental care, and widespread suffering of children from untreated dental caries in low-income populations in Latin America and Asia [18,46,47,48,49,50]. This study also supports the findings of systematic reviews showing that larger family size and greater numbers of children are generally associated with greater risk of ECC [35,36]. In addition, Lam et al., found an increase in caries increment associated with being born as a second-child or higher birth order and having siblings with caries [36]. Similarly, Hooley et al. found greater risk for ECC for children of higher birth order (i.e., having older siblings), as well as those eating more junk food, although the findings were not universal [35]. A study of 5-year-old children in India found dental caries associated with a higher number of siblings as well as parental laxness about child tooth brushing, unassisted brushing by the child, high snacking frequency, and parental inability to limit sugar consumption [51]. A study of school-age children in Nigeria found that having zero to two siblings significantly reduced the odds of caries compared with children with three or more siblings [52]; however, another study of school-age children in Nigeria found that while the last child of the family had increased odds of dental caries, children from larger families had reduced odds of dental caries [53].

In addition to family size and number of children, studies have examined other aspects of family structure and functioning that may affect childhood caries. A study of 5-year-olds in Hong Kong found higher dmft scores associated with having grandparents as their primary caregivers, as well as eating sugary snacks more than twice daily [54]. A U.S.-wide study of children from age 0 to 17 years found that children in grandparent-led families had increased prevalence of oral health problems as well as other physical and mental health problems, and reduced healthcare utilization [55]. A qualitative study from the Appalachian region of the U.S. found that mothers attributed their children’s ECC risk to frequent conflict with partners and grandparents about following recommended feeding and oral hygiene practices [56].

There is clearly a complex set of social determinants and family factors that contribute to parents’ ability to promote their children’s oral health and prevent caries, with some differences and some similarities across different regions and different socioeconomic and family contexts. Our study may add to the understanding of the relationship between family structure and ECC by elucidating some of the intermediary oral health practices that may explain the connection between number of children and ECC; and further underscore the complexity of the relationship, as we found that larger families had both protective factors (less unhealthy bottle feeding) and risk factors (poorer oral hygiene practices). We hypothesize that children in larger families in our sample may have had greater support for exclusive breastfeeding from their family, community and cultural background; however, they may have had fewer financial resources for oral hygiene products and less free time to care for each child’s teeth daily. We did not find that smaller vs. larger families had significantly different practices around limiting sweets or consuming junk food daily; we hypothesize that there may be some healthier and some less-healthy dietary practices that cancel out each other’s effect: in bigger families, some parents may give children junk food to distract or pacify them when the parents are busy, but some may have less disposable money to buy junk food. In smaller families, some parents may spoil their only-children with more junk food, while some may have more time and patience to limit junk food. Overall consumption of sweets was very common in our sample, with one-third of children fed sugary liquids in the baby bottle and over one-third consuming junk food or sugary drinks daily. While the World Health Organization (WHO) guidelines on sugar intake recommend no added sugar consumption under age 2 and afterwards less than 5% of intake from free sugars, the majority of children worldwide and in LMICs—particularly in Latin America and Asia—consume added sugars earlier and in greater quantity and frequency than recommended, contributing to dental caries as well as obesity, type 2 diabetes, cardiovascular disease and cancers [57,58,59]. This suggests a critical need for healthier food environments and oral health and nutrition literacy for families [60,61]. Overall, we found that children in larger families were less likely to be cavity-free, but more likely to have had a dental visit, which we believe is consistent with our findings from a separate analysis of this sample that families primarily seek dental care for dental problems (usually dental pain due to caries) rather than for disease prevention [62].

Other studies have highlighted the connections between family stressors and early childhood caries. Seow developed a conceptual model identifying socio-economic and family stressors that contribute to less-healthy parenting practices and increase children’s risk for ECC [63]. On the positive side, a US-wide study of school-age children found that children living in families with higher levels of resilience and social connection had lower odds of reported dental caries [64]; and a study of preschool-age children of mothers with severe dental caries found that the mother’s perception of greater social support was associated with lower odds of their children experiencing severe caries [65].

Our findings, and those of other studies, suggest the need for a broad range of interventions at the child, family, community, societal and global level to prevent childhood caries and improve children’s oral health and well-being. While numerous studies have identified the risk factors for ECC, it is critical to identify factors that are not simply associated with the environment but rather those that adversely impact family functioning and health behaviors, and that could be addressed by effective interventions. Focusing on oral health, there is a need to expand nutrition and oral health education for children and families, oral health promotion in childcare and schools, and access to dental care for all children and adults. To support oral health and overall health more broadly, there is a need to expand national and local policies for basic family income, universal access to nutritious foods and beverages (including clean water), free education for children and adults, and universal health care including dental care, family planning and the full range of reproductive healthcare services. There is also a need for greater support for healthy parenting through parenting education, paid parental leave from work, affordable childcare, and limits on the marketing of unhealthy products to children and families such as sugary beverages and foods, and baby formula. Finally, there is a need for research to study the impact of these interventions on child oral health, overall health and family wellbeing. Many of these interventions and calls for further research have been recommended by the World Health Organization, FDI World Dental Federation, and the Lancet Commission on Oral Health [22,66,67,68,69].

This study contributes to the literature by helping to elucidate the relationships between number of children in the family, maternal-child oral health practices, and childhood caries in an LMIC population. This study has many limitations: The cross-sectional design cannot prove causality, and the convenience sampling method may limit the generalizability of the findings. Survey responses may be susceptible to recall and social desirability bias. There may have been clustering at family and community levels for which this analysis did not account; and a mediation analysis of the relationship among family size, oral health practices and child caries outcomes could help clarify the findings. The data were collected from 2011 to 2013, and it is unknown whether they are representative of current findings from these sites; however, recent studies in LMICs have similar findings regarding oral health practices and child caries experience, and their associations with the number of children in families. Strengths of the study are that sampling from different countries demonstrates some differences as well as similarities—that greater levels of economic stress caused by additional children in the family may lead to greater child disease burden—and that there is a universal need for greater family support to promote child health and well-being. The similar findings from the three countries that are heterogeneous with respect to their cultures, government, and family structure/characteristics suggest potential generalizability of the results of this study.

## 5. Conclusions

This cross-sectional study of a convenience sample of children aged 6 months through 6 years across 3 LMICs—with a high prevalence and severity of childhood caries and widespread dietary and oral hygiene risk factors–found a significant association between larger family size and some oral health risk and protective practices; and that a one-unit increase in the number of children in the family was significantly associated with a greater likelihood of a child having any decayed, missing or filled teeth (dmft), a greater number of decayed teeth among children with decay, and less likelihood of a child being cavity-free. There is a need for policy interventions to support healthier parenting practices, including expanded oral health education and services as well as universal health care including reproductive health services, basic income and nutritious foods, paid parental leave from work, affordable childcare, and limits on the marketing of unhealthy products to children and families.

## Figures and Tables

**Figure 1 ijerph-20-02195-f001:**
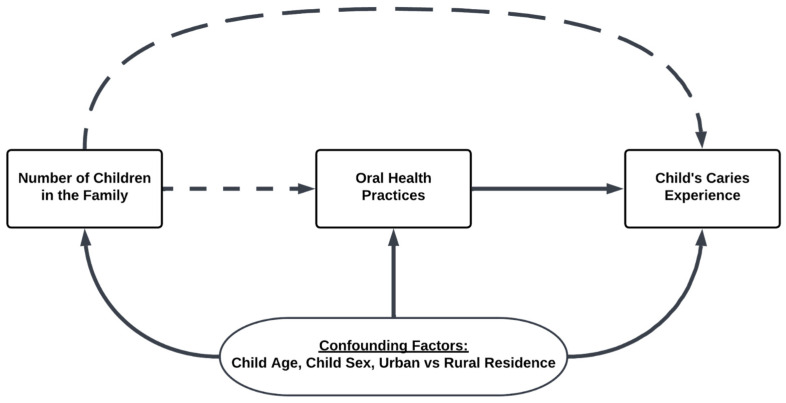
Hypothesized association between the number of children in the family, maternal-child oral health practices and child oral health outcomes. *Dotted: our pathways of interest*.

**Table 1 ijerph-20-02195-t001:** Sociodemographic characteristics of children and families.

Characteristics	Total N = 1352 Children/ Mothers ^a^	Ecuador N = 290 Children/ Mothers ^a^	Nepal N = 469 Children/ Mothers ^a^	Vietnam N = 593 Children/ Mothers ^a^	One Child N = 496 Children/ Mothers ^a^	Multiple Children N = 851 Children/ Mothers ^a^	Difference between One Child and Multiple Children (*p*-Value) ^b^
Site Type	Urban: 51.7% Rural: 48.3%	Rural: 100%	Urban: 69.7% Rural: 30.3%	Urban: 100%	Urban: 83.1% Rural: 16.9%	Urban: 57.9% Rural: 43.0%	<0.001
**Children**							
Age (years)	3.7 ± 1.5	3.8 ± 1.8	3.7 ± 1.7	3.6 ± 1.0	3.2 ± 1.4	4.0 ± 1.4	<0.001
Sex (Male/Female)	52.7%/47.3%	54.8%/45.2%	53.2%/46.8%	51.2%/48.8%	55.9%/44.1%	50.2%/49.8%	0.076
**Mothers and Households**							
Age (years)	30.9 ± 6.4	30.4 ± 8.4	28.6 ± 5.4	33.0 ± 5.1	28.7 ± 5.7	32.2 ± 6.4	<0.001
Education (years)	8.7 ± 5.5	7.1 ± 4.8	5.1 ± 4.4	12.8 ± 3.7	10.9 ± 4.8	7.4 ± 5.5	<0.001
Number of Children	2.2 ± 1.6	3.7 ± 2.4	2.2 ± 1.1	1.6 ± 0.7	1.0 ± 0.0	2.9 ± 1.6	<0.001
Range of Number of Children	1–12	1–12	1–7	1–5	1	2–12	
Number in Household	5.3 ± 2.3	7.0 ± 2.9	5.1 ± 2.0	4.5 ± 1.7	4.6 ± 2.2	5.7 ± 2.3	<0.001
Potable Water at Home	79.3%	32.1%	85.5%	98.6%	89.1%	73.6%	<0.001
Electricity	98.0%	87.6%	98.5%	99.5%	98.6%	95.4%	<0.001
Cooking Fuel Other Than Wood (Gas or Electric)	84.1%	86.9%	63.7%	98.8%	92.5%	79.0%	0.019
**Time to walk from home to a store that sells junk food:**							
Less than 5 min	84.7%	64.4%	78.9%	82.3%	85.1%	73.3%	<0.001
6–20 min	16.4%	23.4%	13.9%	15.5%	12.8%	23.5%	<0.001
Over 20 min	5.9%	12.1%	8.1%	2.2%	2.1%	3.2%	0.221

^a^ Data is represented as mean ± SD or %. ^b^ Student *t*-test: *p*-value < 0.05 is significant.

**Table 2 ijerph-20-02195-t002:** Maternal-child oral health and nutrition knowledge and practices ^a^.

Characteristics	Total N = 1352 Children/ Mothers ^a^	Ecuador N = 290 Children/ Mothers ^a^	Nepal N = 469 Children/ Mothers ^a^	Vietnam N = 593 Children/ Mothers ^a^	One Child N = 496 Children/ Mothers ^a^	Multiple Children N = 851 Children/ Mothers ^a^	Difference between One Child and Multiple Children (*p*-Value) ^b^
**Mother’s knowledge of the causes of childhood caries ^c^**							
Don’t know	15.1%	14.8%	10.9%	20.0%	14.7%	15.4%	<0.001
Eating sweets	73.6%	66.2%	85.3%	66.8%	73.8%	73.5%	0.891
Not brushing teeth	26.4%	44.8%	27.7%	16.4%	24.4%	27.6%	0.196
Drinking soda/juice	3.8%	6.6%	5.3%	1.2%	3.6%	3.9%	0.768
Using the baby bottle	0.9%	3.8%	0.2%	0.0%	0.2%	1.3%	0.040
**Mother’s knowledge of the complications of childhood caries ^c^**							
Don’t know	5.5%	11.4%	7.2%	0.2%	3.8%	6.5%	0.046
Pain	56.0%	62.8%	76.8%	31.1%	49.3%	59.9%	<0.001
Difficulty eating	40.9%	27.9%	45.0%	44.9%	41.9%	40.4%	0.591
Difficulty sleeping	14.6%	19.3%	17.5%	8.9%	15.2%	16.1%	0.056
Harm their health	22.3%	13.1%	8.7%	41.3%	26.7%	19.7%	0.004
**Child Nutrition Practices**							
Child breastfed	97.3%	97.5%	98.7%	96.0%	96.1%	98.1%	0.020
Mean duration of breastfeeding (months)	18.3 ± 12.2	17.7 ± 8.2	22.5 ± 16.0	14.9 ± 8.4	19.0 ± 12.1	22.5 ± 13.5	<0.001
Child bottle fed (mixed breast- and bottle feeding)	52.8%	40.1%	19.0%	86.1%	64.2%	46.0%	<0.001
Mean duration of bottle feeding (months)	27.5 ± 15.2	21.2 ± 15.2	20.3 ± 14.4	30.1 ± 14.5	29.4 ± 15.1	27.6 ± 15.0	0.014
Unhealthy bottle feeding practices	44.6%	35.6%	13.2%	74.3%	54.6%	38.6%	<0.001
Bottle fed with sugary liquid	29.4%	31.7%	1.9%	62.3%	35.8%	25.8%	<0.001
Bottle fed while sleeping	14.6%	12.0%	7.1%	22.1%	16.2%	13.5%	0.184
Bottle fed past 24 months of age	34.3%	18.7%	8.3%	62.8%	43.6%	28.7%	<0.001
Daily consumption of milk	67.2%	20.8%	69.9%	91.0%	79.7%	60.1%	<0.001
Daily consumption of soda	6.1%	7.8%	5.0%	5.9%	6.3%	6.0%	0.847
Daily consumption of candy/sweets	29.9%	21.8%	53.4%	12.9%	26.0%	32.2%	0.024
Daily consumption of chips/biscuits	17.6%	15.9%	25.6%	11.6%	17.6%	17.7%	0.954
Daily consumption of any junk food or sugary drink	34.6%	28.9%	56.0%	18.2%	36.5%	40.4%	0.048
Mother calms fussy child with sweets	13.8%	9.7%	27.5%	1.8%	11.3%	15.2%	0.058
Mother restricts candy	6.9%	7.9%	13.6%	1.0%	6.0%	7.4%	0.333
**Child Oral Health Practices**							
Child has own toothbrush	88.2%	90.8%	75.7%	97.0%	88.4%	88.0%	0.810
Child has toothpaste	90.4%	84.4%	89.6%	94.1%	91.9%	89.6%	0.169
Mother helps with brushing frequently/almost always	54.6%	54.0%	24.5%	80.0%	64.2%	48.9%	<0.001
Mother never/occasionally helps with brushing	45.4%	46.0%	75.5%	20.0%	35.8%	51.1%	<0.001
Mother does nothing to care for child’s teeth	6.4%	7.6%	12.6%	0.2%	5.3%	7.1%	0.223
Child has visited the dentist	38.1%	74.0%	9.7%	44.5%	30.9%	42.0%	<0.001

^a^ Data is represented as mean ± SD or %. ^b^ Student *t*-test: *p*-value < 0.05 is significant. ^c^ Interview question was phrased as open-ended questions.

**Table 3 ijerph-20-02195-t003:** Child dental status ^a^.

Characteristics	Total N = 1352 Children/ Mothers ^a^	Ecuador N = 290 Children/ Mothers ^a^	Nepal N = 469 Children/ Mothers ^a^	Vietnam N = 593 Children/ Mothers ^a^	One Child N = 496 Children/ Mothers ^a^	Multiple Children N = 851 Children/ Mothers ^a^	Difference between One Child and Multiple Children (*p*-Value) ^b^
**Child Tooth Decay**							
Child has any decayed, missing or filled teeth (dmft)	72.3%	86.9%	59.9%	75.0%	64.1%	77.2%	<0.001
Range in number of dmft	0–24	0–24	0–17	0–23	0–20	0–24	
Mean dmft	5.4 ± 5.3	8.3 ± 5.4	3.1 ± 3.8	5.9 ± 5.4	4.4 ± 4.9	6.0 ± 5.4	<0.001
Mean # dmft for children with caries	7.5 ± 4.8	9.6 ± 4.6	5.2 ± 3.6	7.8 ± 4.9	6.9 ± 4.5	7.8 ± 4.9	0.003
Prevalence of deep decay, in or near pulp	28.5%	39.7%	20.7%	31.2%	16.9%	28.0%	<0.001
**Child Mouth Pain**							
Any mouth pain (occasionally/frequently/always)	51.2%	47.2%	27.5%	71.8%	52.4%	50.4%	0.023
Mouth pain (frequently/always)	22.9%	18.1%	5.5%	39.0%	26.9%	20.4%	0.617
**Mother’s Overall Assessment of Child’s Health**							
Mother’s assessment of child’s oral health as “bad”	17.4%	13.8%	21.9%	15.5%	15.1%	18.8%	0.089
Mother’s assessment of child’s overall health as “bad”	2.9%	3.9%	3.2%	2.1%	2.3%	3.1%	0.365

^a^ Data is represented as mean ± SD or %. ^b^ Student *t*-test: *p*-value < 0.05 is significant.

**Table 4 ijerph-20-02195-t004:** Association between number of children, oral health practices, and caries experience ^a^.

Logistic Regression Analysis				
Oral Health Practices	Unadjusted Odds Ratio (OR) (95% Confidence Interval)		Adjusted Odds Ratio (aOR) ^a^ (95% Confidence Interval)	Significance of aOR *p*-Value
**Nutrition practices**				
Unhealthy baby bottle feeding	0.75 (0.69–0.82)		0.85 (0.77–0.93)	0.0009
Daily consumption of junk food	0.97 (0.90–1.04)		1.00 (0.92–1.09)	0.96
Calms child with sweets	1.04 (0.94–1.13)		0.99 (0.88–1.11)	0.93
Mother restricts candy	1.08 (0.96–1.21)		1.03 (0.89–1.18)	0.65
**Oral Hygiene Practices**				
Child has own toothbrush	0.90 (0.82–0.99)		0.89 (0.80–0.99)	0.031
Child has toothpaste	0.80 (0.74–0.88)		0.83 (0.75–0.93)	0.0007
Mother helps child brush	0.78 (0.72–0.85)		0.86 (0.79–0.94)	0.001
Child has been to the dentist	1.33 (1.23–1.45)		1.19 (1.09–1.30)	0.0001
**Child Caries Experience**				
Child has any decayed, missing or filled teeth (dmft)	1.36 (1.22–1.52)		1.18 (1.06–1.32)	0.005
**Zero-Inflated Poisson (ZIP) Regression Analysis**				
**Oral Health Outcome**	**Count Model** **Estimate ^b,d^** **(95% CI)**	**Count Model** ***p*-value**	**Inflation Model** **Estimate ^c,d^** **(95% CI)**	**Inflation Model** ***p*-value**
Number of decayed, missing and filled teeth (dmft)	1.04 (1.01–1.06)	<0.0001	0.85 (0.76–0.94)	0.005

^a^ Primary predictor: number of children as a continuous variable; adjusted for child age, child sex, and urban/rural site type, reference category is the absence of the stated characteristic, e.g., reference category for unhealthy baby bottle feeding is breastfeeding only or bottle feeding without sugary liquids, not bottle feeding to sleep, and bottle feeding for less than 24 months. ^b^ Expected dmft count, among children with any decayed, missing and filled teeth; baseline estimate: 4.57. ^c^ Odds of being in the zero dmft group (i.e., being cavity-free), among all children. ^d^ Percentile-based confidence interval.

## Data Availability

The original datasets analyzed in this study are not publicly available in accordance with participant privacy, informed consent forms that did not include release of the data, and the study’s approved IRB protocols. However, the minimal dataset necessary to interpret, replicate and build upon the findings of this study are available from the senior author [KS-G] upon reasonable request.

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
