# Peer review of "Fewer Children in Families Associated with Lower Odds of Early Childhood Caries: A Sample from Three Countries"

_ijerph, 2023, doi:10.3390/ijerph20032195_

Round 1
Reviewer 1 Report
“Fewer Children in Families Associated with Lower Odds of Early Childhood Caries: A Sample from 3 Countries” is an interesting paper aimed to explore the relationship between families' number of children, oral health practices and child caries experience in a convenience sample of 1,374 children aged 6 months through 6 years and their families from Ecuador, Nepal, and Vietnam.
I am interested to know more about study sites (regions or provinces or municipalities) from all the countries. As the authors have included a convenience sample of 1,374 low-income mothers and children aged 6 months through 6 years participating in a family 72 of community-based programs, how did authors conclude on low/high income when the questionnaire did not include family income? In addition, why did authors use urban vs rural residence as a proxy measure to categorise the SES during analyses when the sample represented the low-income population? Did the authors compute the sample size for this study? Examiners were trained and calibrated, can authors present the agreement level? As the explanatory variable did not include weight or BMI, is it worth mentioning in the methods (Page 3 lines 101-102)? This may confuse the readers.
Why did the authors perform both logistic regression and Zero-Inflated Poisson regression analyses to assess the association between outcome and explanatory variables? How was the outcome variable dichotomized for the logistic regression? The study's aim was to explore the relationship between families' number of children, oral health practices and child caries experience. It would be worthwhile to present the estimates for covariates used in the ZIP model.
I suggest authors review the caption of Table 3 as “Child dental status”.
Authors could discuss more on the patterns of introducing sugar-containing foods/drinks at an early age in the LMCIs, particularly Ecuador, Nepal, and Vietnam. Moreover, the generalisability of the study findings must be discussed.
Reviewer 2 Report
This study aims to to explore the association between the number of children in a family, oral health practices, and children’s experience 67 of ECC by their number of decayed, missing, or filled teeth (dmft).
The topic of this manuscript is very interesting and is well-written. I have the following comments and suggestions:
1. Keywords: please review if all terms are MeSH terms
2. Please be sure that the manuscript accomplishes the strobe guidelines for observational studies (please attached the file with the checklist)
3. The study was conducted in 2011-2013 and several conditions have passed since this study period, which constitutes a limitation. Please mention this in the discussion section.
4. I would like to know if regional differences exist in the multivariate models conducted.
5. Please mention more deeply in the discussion recommendations for research and practice based on findings.
Author Response
Reviewers Comments/Feedback
Thank you so much for your comments and suggestions. Your insights and feedback have strengthened the quality of the paper, and we are very grateful. Please find our responses to each comment below. We have attempted to carefully address each comment in the responses and corresponding edits and additions made to the manuscript text and tables. As requested by the journal editor, all changes in the manuscript were made using the “Track changes” feature.
Reviewer 2:
Comment 1: Please be sure that the manuscript accomplishes the strobe guidelines for observational studies (please attached the file with the checklist)
Response 1: Thank you for your comment. Please find the attached STROBE Checklist for cross-sectional studies (https://www.equator-network.org/reporting-guidelines/strobe/) below.
Comment 2: The study was conducted in 2011-2013 and several conditions have passed since this study period, which constitutes a limitation. Please mention this in the discussion section.
Response 2: Thank you for your comment. We added a sentence addressing this in the limitations section of the Discussion.
Comment 3:
More analysis is presented on this in our updated Supplementary Section of our manuscript.
Comment 4: Keywords: please review if all terms are MeSH terms
Response 4: Thank you for your comment. Terms were reviewed using MeSH, and non-MeSH terms were removed or modified.
Comment 5: Please mention more deeply in the discussion recommendations for research and practice based on findings.
Response 5: Thank you for your comment. We expanded recommendations for research and practice in the Discussion section.

Reviewer 3 Report
A review report of the manuscript titled “Fewer Children in Families Associated with Lower Odds of Early Childhood Caries: A Sample from 3 Countries”. Authors of current paper aimed to explore the relationship between families' number of children, oral health practices and child caries experience in a convenience sample of 1,374 children aged 6 months through 6 years and their families from Ecuador, Nepal, and Vietnam. They concluded that early childhood oral health promotion should include focus on oral hygiene and healthy feeding––particularly breastfeeding and healthy bottle feeding practices––as well as access to family planning services and support for childcare.
Here are my concerns, questions and recommendations:
1. Was there any null hypothesis? Was is confirmed/rejected?
2. If data were collected from 2011 to 2013, why authors decided to publish it in 2023? Are these date valid nowadays?
3. Please provide questionnaire as a supplementary material.
4. Provide the sample size calculation.
5. Methods section should be expanded and more information should be added with appropriate citations.
6. I recommend to add the following paper in the discussion:
https://opendentistryjournal.com/VOLUME/16/ELOCATOR/e187421062209270/
Author Response
Reviewers Comments/Feedback
Thank you so much for your comments and suggestions. Your insights and feedback have strengthened the quality of the paper, and we are very grateful. Please find our responses to each comment below. We have attempted to carefully address each comment in the responses and corresponding edits and additions made to the manuscript text and tables. As requested by the journal editor, all changes in the manuscript were made using the “Track changes” feature.
Reviewer 3:
Comment 1: Was there any null hypothesis? Was is confirmed/rejected?
Response 1: Thank you for your comment. Concerning the t-test, the null hypothesis µ1 − µ2 = 0 calculated by the p-value in Table 2 and 3 was rejected for some of the relevant categories in Table 4 which include: Unhealthy Baby Bottle Feeding, Daily Consumption of Junk Food, Mother Helps Child Brush, Child has been to the Dentist, and Child has any dmft.
Comment 2: If data were collected from 2011 to 2013, why authors decided to publish it in 2023? Are these date valid nowadays?
Response 2: Thank you for your comment. We have previously published descriptive analyses from individual countries in this dataset, and have progressed over time to examine associations between factors as well as similarities and differences across the countries. We do not know whether the data from 2011-2013 is representative of current findings from these sites. However, recent studies in LMICs and these 3 countries have similar findings regarding oral health practices and child caries experience, as well as associations with the number of children in families.
Comment 3: Please provide questionnaire as a supplementary material.
Response 3: We have included the questionnaires below.
Comment 4: Provide the sample size calculation.
Response 4: This study was based on convenience sampling, and no sample size calculation was performed prior to data collection. However, we have now included an estimate of implied power based on the acquired sample size and observed differences. Implied power analysis was conducted using G*Power which suggested a 95% certainty that the observed findings were due to a real difference in subgroups.
Comment 5: Methods section should be expanded and more information should be added with appropriate citations.
Response 5: Thank you for your comment. We have expanded the Methods section with citations.
Comment 6: I recommend to add the following paper in the discussion:
https://opendentistryjournal.com/VOLUME/16/ELOCATOR/e187421062209270/
Response 6: Thank you for this recommendation. The suggested paper was added to the Discussion section.
